META-RESEARCH ARTICLE

# Statistical simulations show that scientists need not increase overall sample size by default when including both sexes in in vivo studies

**Benjamin Phillips**[1], **Timo N. Haschler**[2], **Natasha A. Karp**[1]*

**1** Data Sciences & Quantitative Biology, Discovery Sciences, R&D, AstraZeneca, Cambridge, United Kingdom, **2** Bioscience Renal, Research and Early Development Cardiovascular, Renal and Metabolism, Biopharmaceutical R&D, AstraZeneca, Cambridge, United Kingdom

* natasha.karp@astrazeneca.com

## Abstract

In recent years, there has been a strong drive to improve the inclusion of animals of both sexes in the design of in vivo research studies, driven by a need to increase sex representation in fundamental biology and drug development. This has resulted in inclusion mandates by funding bodies and journals, alongside numerous published manuscripts highlighting the issue and providing guidance to scientists. However, progress is slow and barriers to the routine use of both sexes remain. A frequent, major concern is the perceived need for a higher overall sample size to achieve an equivalent level of statistical power, which would result in an increased ethical and resource burden. This perception arises from either the belief that sex inclusion will increase variability in the data (either through a baseline difference or a treatment effect that depends on sex), thus reducing the sensitivity of statistical tests, or from misapprehensions about the correct way to analyse the data, including disaggregation or pooling by sex. Here, we conduct an in-depth examination of the consequences of including both sexes on statistical power. We performed simulations by constructing artificial datasets that encompass a range of outcomes that may occur in studies studying a treatment effect in the context of both sexes. This includes both baseline sex differences and situations in which the size of the treatment effect depends on sex in both the same and opposite directions. The data were then analysed using either a factorial analysis approach, which is appropriate for the design, or a *t* test approach following pooling or disaggregation of the data, which are common but erroneous strategies. The results demonstrate that there is no loss of power to detect treatment effects when splitting the sample size across sexes in most scenarios, providing that the data are analysed using an appropriate factorial analysis method (e.g., two-way ANOVA). In the rare situations where power is lost, the benefit of understanding the role of sex outweighs the power considerations. Additionally, use of the inappropriate analysis pipelines results in a loss of statistical power. Therefore, we recommend analysing data collected from both sexes using factorial analysis and splitting the sample size across male and female mice as a standard strategy.

**Data Availability Statement:** The R scripts used to generate the data and figures and analyse the in vivo data have been made available as a Zenodo

repository (https://doi.org/10.5281/zenodo.7806724).

**Funding:** The authors received no specific funding for this work.

**Competing interests:** I have read the journal's policy and the authors of this manuscript have the following competing interests: BP, TH, and NAK have shareholdings in AstraZeneca.

## Introduction

There has been a bias towards using a single sex in in vivo research. Though there is variation between subdisciplines, this strategy has tended to result in a heavy bias towards male animals. For example, in 2009, only 26% of studies used both sexes and among the remainder there was a male bias in 80% of studies [1]. The negative consequences of these shortcomings on scientific enterprise are beginning to be better understood as evidence emerges that our current fundamental biological knowledge base may be biased. For example, a recent report concluded that the fundamental molecular basis of pain is highly sex dimorphic, yet much of our knowledge in this area is derived from studies solely using male organisms [2]. This situation risks generating a knowledge imbalance that persists through the research pipeline, ultimately manifesting in the clinic.

To improve the translation of results from animals to humans, there has been a push to include both male and female animals in in vivo studies. For example, numerous funding bodies, including the NIH in the United States and the MRC in the United Kingdom, now have inclusion mandates. These policies do not require scientists to study differences between males and females per se, but instead aim to improve the generalisability of studies by calculating an average effect estimated from both sexes. If, however, there is a large, meaningful sex difference in the treatment effect, studies should be designed in such a way that the visualisation and analysis detects it [3,4]. The NIH policy introduced the term Sex as a Biological Variable (SABV), and, here, we use the term to represent a sex inclusive research philosophy that emphasises the importance of automatic inclusion, with a focus on treatment effect estimates. Any of a wide range of factors including animal strain, age, health status, or others could also be the focus of a campaign to improve research generalisability. However, sex is a particularly pressing and timely direction for improved representation since females account for such a large proportion of the population of interest but are currently largely overlooked.

Over time, the proportion of studies including both sexes has improved [5,6], with one study estimating an increase between 2009 and 2019 from 26% to 48% of studies [6]. Scientists tend to be supportive of efforts to improve sex representation in in vivo research [7]. Unfortunately, in studies where both sexes are tested, a large proportion commit errors at the statistical analysis stage [8]. Thus, despite an overall increase in inclusion, the proportion of studies appropriately interpreting the influence of sex is still low [5]. Overall, the pace of change is slow, owing to a persistent and broad range of perceived statistical and practical barriers. Consequently, scientists believe that including both sexes will introduce a significant ethical, practical, and financial burden [9]. The barriers include now debunked beliefs that female animals produce more variable data [10–12], institution-level ingrained cultural belief about the value of studying 2 sexes [13,14], and a skill-gap in handling data collected under factorial designs [8]. There is also a general belief that it is necessary to greatly increase the experimental sample size (N) when investigating treatment effects in 2 sexes [14,15]. For example, a recent report cites that 27% of published papers justified a single sex approach due to concerns around experimental variability [6]. Though this misconception has been addressed previously [11,16], and guidance on appropriate analysis exists [17,18], it remains widespread, and there is a need for a deeper exploration of the impact of including both sexes on statistical power. Revisiting this is critical to enable the community to address this significant barrier to sex-inclusive research due to the misguided belief that there is a trade-off between pursuing the 3Rs (replacement, reduction, refinement) by means of reducing animal usage on the one hand and designing more generalisable studies on the other [14].

When considering sex-inclusive research, the following misconceptions and data analysis errors have been reported:

- Misconception 1: Designs that include both sexes will require a doubling of sample size to achieve the same power [9,14,19].

- Misconception 2: Belief that the possibility of sex effects (either a baseline differences or a treatment effect that depends on sex) will increase variability and consequently require an increased N to maintain the power [11,12].

- Error 1: Inappropriate pooling of male and female data for a treatment group (i.e., combining the data from both sexes and ignoring sex as a factor in the analysis) [8].

- Error 2: Disaggregation of the data by sex and independent statistical comparison between the control and treated group. Then, comparing the *p*-values from the independent tests [8,20].

- Error 3: Incorrect groups in statistical comparison: comparison of treated males and treated females [6].

Of these, the misconceptions (1 and 2) contain empirical claims, and for these we have constructed a range of simulated datasets to extensively test statistical power for a range of plausible biological scenarios where both sexes are tested and subsequently analysed by an appropriate factorial pipeline. The consequences of pooling male and female data and disaggregating the data by sex (error numbers 1 and 2) are additionally demonstrated as part of these simulations. A case study example analysis (S1) explores error 3.

Through simulations, we have conducted an in-depth examination of the impact of sex inclusion on statistical power for a variety of commonly implemented analysis strategies. Our methodology has been designed to demonstrate the problems that result from using the wrong analysis strategies and address common misconceptions around power when using factorial designs and analysis. Our results demonstrate that including both male and female animals does not reduce statistical power across a wide range of outcomes when investigating a treatment effect. We demonstrate that power loss is (a) rare and (b) indicative of a sex dimorphism that it would be important to be aware of. Our comparison of statistical analysis strategies demonstrates that inappropriate methods, including both pooling the data from males and females and disaggregation of the data by sex result in a loss of statistical power. Therefore, the importance of adopting a factorial analysis method is central to appropriately analyse data from studies testing a treatment effect in both sexes. To support the adoption of appropriate analysis, we provide a case study (S1 Case Study) demonstrating an example pipeline for analysing data collected under such designs, intended as a practical guide for scientists. A compendium of common statistical terms used within this manuscript is also included as a guide to readers (see Box 1 Glossary).

---

**Box 1.** Glossary.

Common statistical terms used within this manuscript, adapted from [28] and placed in the context of in vivo research are detailed below:

**Effect size**: Quantitative measure of differences between groups or strength of relationships between variables.

**Factor**: Factors are independent categorical variables that the experimenter controls during an experiment in order to determine their effect on the outcome variable. Example factors include sex or treatment.

---

**Factorial design**: An experimental design that is used to study 2 or more factors, each with multiple discrete possible values or levels.

**Independent variable**: A variable that either the experimenter controls (e.g., treatment given or dose) or is a property of the sample (sex) or a technical feature (e.g., batch or cage) that can potentially affect the outcome variable.

**Interaction effect:** When the effect of one independent variable (factor) depends on the level of another. For example, the observed treatment effect depends on the sex of the animals.

**Levels**: Are the values that the factor can take. For example, for the factor sex the levels are male and female.

**Main effect**: A main effect is the overall effect of one independent variable on the outcome variable averaging across the levels of the other independent variable.

**Outcome variable**: A variable captured during a study to assess the effects of a treatment. Also known as dependent variable or response variable.

**Power**: For a predefined, biologically meaningful effect size, the probability that the statistical test will detect the effect if it exists (i.e., the null hypothesis is rejected correctly). Can also be called sensitivity.

**Treatment**: A process or action that is the focus of the experiment. For example, a drug treatment or a genetic modification.

**Sex as a biological variable (SABV):** The research philosophy that emphasises the importance of including both sexes in in vivo studies in such a way that a generalisable treatment effect is detectable. Critically, sex should be treated as a variable of primary biological interest. There is no requirement to prospectively power a study to detect a baseline difference between the sexes or treatment by sex interaction, but studies will detect large differences where they exist.

## Results and discussion

### Introduction to simulations to explore the impact of sex-related biological differences on statistical power

It is often believed that the inclusion of both sexes increases the sample size needed to detect an equivalently sized treatment effect. We have systematically investigated this claim by the construction of simulated datasets to represent sex-inclusive designs where sex can introduce variance, either as a baseline difference between the sexes or as a treatment effect that is dependent on sex, to assess the impact on power. The objective of these simulations is to understand general trends and relationships and consequently the exact values are not relevant but represent typical animal experiments. They are constructed to support inclusive designs for research programmes in which sex is not the primary question of interest, but where the researcher seeks to understand the generalisability of a treatment effect, thus enabling inference about the effect on both sexes.

In the following simulations, we construct datasets with defined differences in the size of main effects (sex and treatment) and interactions. By repeating this construction process multiple times and testing the resultant data, we can calculate the statistical power of each scenario by returning

## Power Analysis by Simulation

Power analysis can be achieved by generating artificial datasets with specified differences between groups. In this study, we adopted this approach to examine the impact of altering treatment and sex effects on the power to detect a treatment effect.

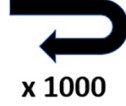

1) Generate simulated data with specified effect size differences between control and treated and each sex.

2) Apply statistical test and extract p values for each statistical comparison of interest.

3) Repeat process 1000 times for each effect size of interest and calculate the % of significant p values as the statistical power for that scenario. Since we know the "true" effect the power is the proportion of times we detect the effect using our statistical test.

**Fig 1. Power analysis by simulation an explainer.**

the proportion of times a statistically significant result occurs (Fig 1). The datasets were analysed with either a factorial pipeline (Fig 2) or a pooled analysis pipeline. The post hoc pairwise statistical tests (e.g., a statistical comparison between untreated and treated animals of each sex), typically implemented when a significant interaction is present, is statistically equivalent to a disaggregation strategy where the data is split by sex and subjected to 2 independent Student's *t* tests.

## Scenario 1: Impact on statistical power of a baseline sex difference

In the first scenario, we tested the effect of increasing baseline sex difference (none = 0, small = 0.5, large = 1 added to the males) on the ability to detect a main effect of treatment (effect size: 0 to 1 added to both sexes) (Fig 3A). This is representative of the most common situation observed in studies run including both sexes [21]. As the effect size of the baseline sex differences increased, there was no change in statistical power when a factorial statistical

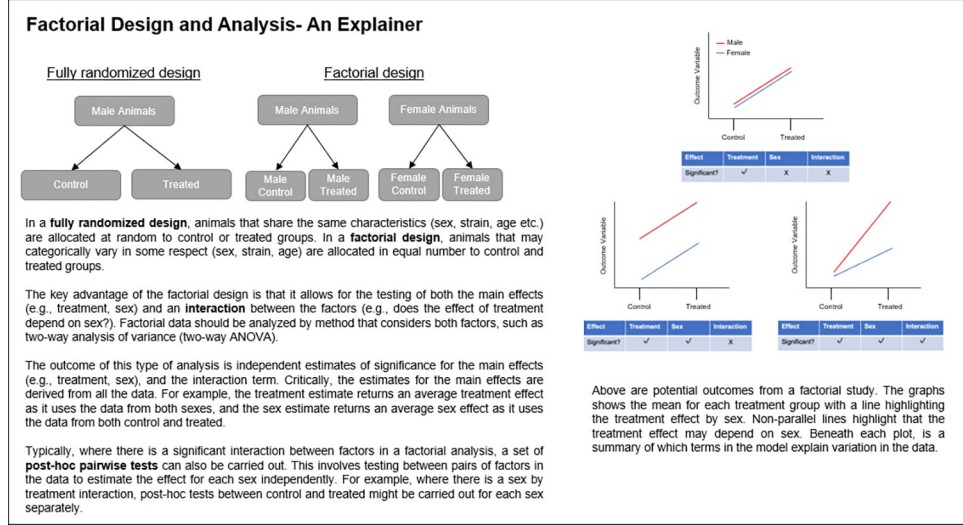

**Fig 2. An explainer on factorial design and analysis.**

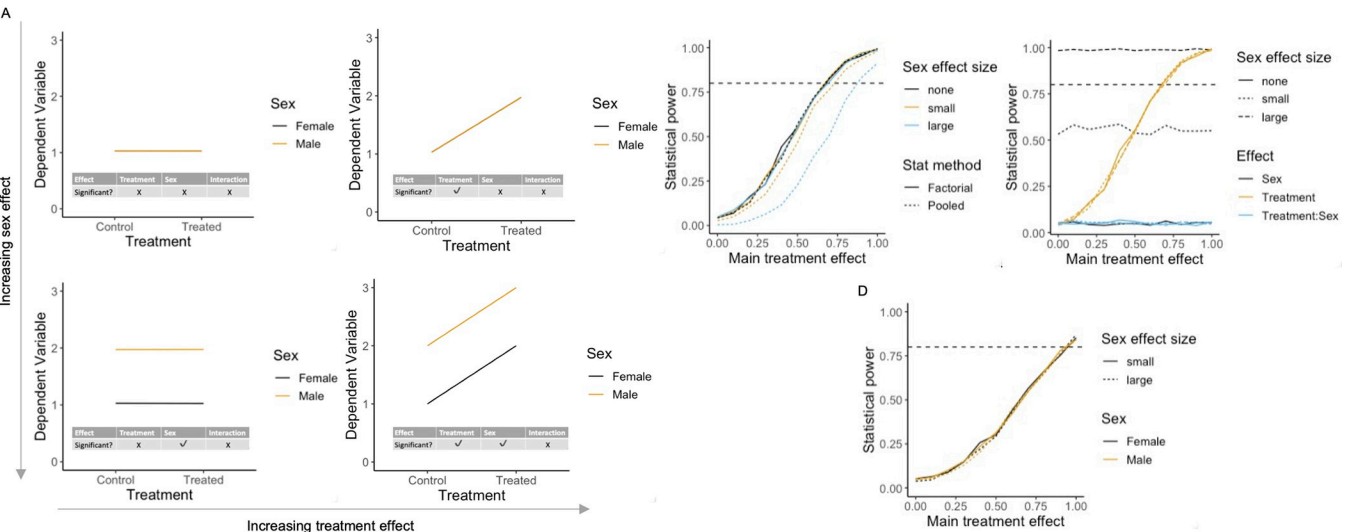

**Fig 3. Results of simulations to explore impact of a baseline sex differences on statistical power.** (A) Illustrative plots of constructed datasets, ranging from zero to maximum baseline sex differences and treatment effect. (B) Statistical power from both a factorial and pooled analysis to detect a main effect of treatment when the main effect of sex varies (none = 0, small = 0.5, large = 1). (C) Power for each model term within factorial analysis output. (D) Power for post hoc comparison of control vs. treated within each sex independently. Simulation $N$ = 1,000 for each scenario assessed. For the graphs of power (B–D), the horizontal dashed line indicates the target power. The data underlying this figure can be found in https://doi.org/10.5281/zenodo.7806724.

method was adopted (Fig 3B). However, there was a loss in power when the data were pooled and a Student's $t$ test applied (error 1) (Fig 3B). This is because factorial analysis is accounting for variability in the data that arose from a baseline difference in the sexes, whereas the effect is not accounted for in the pooled analysis. This highlights the error that is committed when data from both sexes are pooled, resulting in significant loss of power to detect treatment effects.

As the treatment effect was added to both sexes equally no biological interaction occurred. Consequently, there was no power to detect an interaction effect across the range of treatment and sex main effects (Fig 3C). The power to detect a baseline sex difference increased as the effect size increased (Fig 3C). The power for post hoc tests increased equivalently for both sexes as the overall treatment effect increased (Fig 3D). Notably, since the power for the post hoc treatment test is equivalent to the power of analysing the 2 sexes independently, the loss of power when disaggregating the data is evident from comparison of Fig 3D and 3B. This loss of power is a concern because the interaction was not significant, and the treatment effect should be evaluated as a main effect. Furthermore, where there is no baseline sex effect in this scenario, these simulations show that the power to detect a treatment effect is identical between the factorial and pooled strategy (Fig 3B). Often scientists justify the need to pool data to maximise sensitivity arguing that the simpler model would be more powerful when sex does not explain variation. However, these simulations found no statistical cost to using the more complex analysis strategy when there is no variance attributable to sex.

In summary, there is no loss of power to detect a treatment effect when there is a baseline sex effect in the data provided data is not disaggregated.

## Scenario 2: Impact on statistical power when there is a treatment effect of different sizes for the 2 sexes

To understand the impact of an interaction between treatment and sex on the power, where the interaction was in the same direction as a main effect of treatment, datasets were constructed to allow an exploration of the impact of an increasing interaction effect (none = 0,

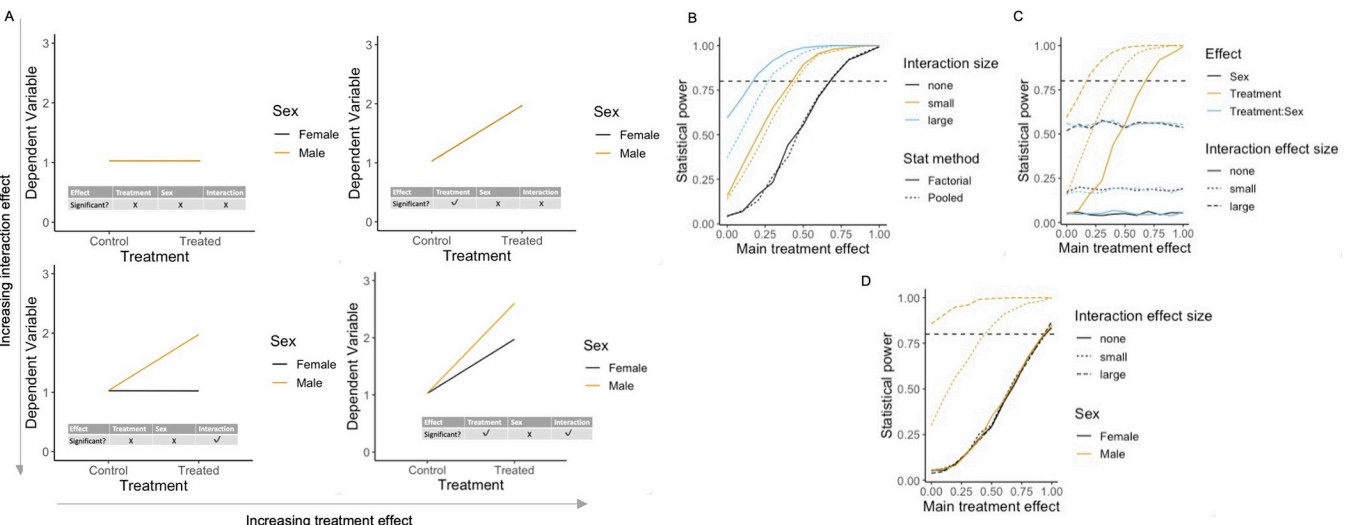

**Fig 4. The impact of a sex-dependent treatment effect on statistical power.** Results of simulations to calculate statistical power where there is an interaction between treatment and sex in the data where the interaction effect is in the same direction as the main treatment effect. (A) Illustrative plots of constructed datasets, ranging from no effect to maximum treatment by sex interaction (none = 0, small = 0.5, large = 1) for varying sizes of a main treatment effect (0–1). (B) Statistical power to detect treatment effect per size of interaction effect and statistical method. (C) Power for each model term within factorial analysis output. (D) Power for post hoc comparison control vs. treated within each sex. Simulation $N$ = 1,000 for each scenario assessed. For the graphs of power (B–D), the horizontal dashed line indicates the target power. The data underlying this figure can be found in https://doi.org/10.5281/zenodo.7806724.

small = 0.5, large = 1) in the presence of a main effect of treatment (varied from 0 to 1) (Fig 4A, all graphs). There was no baseline sex difference in the data. In the factorial pipeline, as the treatment by sex interaction increased (Fig 4A, bottom left and bottom right), the power to detect a main effect of treatment increased in line with the size of the interaction effect (Fig 4B). There was also an increase in power to detect the treatment by sex interaction as the interaction effect size increased (Fig 4C). In the post hoc tests, the power was highest in the sex that exhibited the largest treatment effect, which increased as a function of effect size (Fig 4D).

In the pooled analysis pipeline, as the main treatment effect increased the power increased (Fig 4A, top right and bottom right). However, as the interaction size increased, the pooled statistical pipeline had lower power to detect the main effect of treatment compared to the factorial pipeline (Fig 4B), as there was an increase in sex-related variance not accounted for. This analysis aligns with the belief that sex-related variance reduces statistical power to detect a treatment effect but is arising from inappropriate analysis.

The factorial pipeline first simultaneously tests the main and interaction effects. If the interaction effect is statistically significant and of biological interest, then a subsequent post hoc testing step allows an assessment of treatment effect independently for each sex. If researchers follow a disaggregation pipeline, where data is automatically disaggregated by sex and Student's *t* tests conducted on each separately, there is no ability to assess for a treatment by sex interaction. Often researchers will compare the calls of significance to assess for a differential treatment effect (error 2). However, it is a statistical error to compare 2 independent statistical tests (e.g., a significant and nonsignificant result) and claim a difference on this basis [22–24]. Additionally, disaggregation of the data will reduce the power for the treatment effect (comparison of Fig 4B and 4D). Thus, a factorial analysis is required to properly implement the SABV philosophy and to avoid misinterpretation of the data.

In summary, a factorial pipeline is essential to maintain power and support a claim of a differential sex effect.

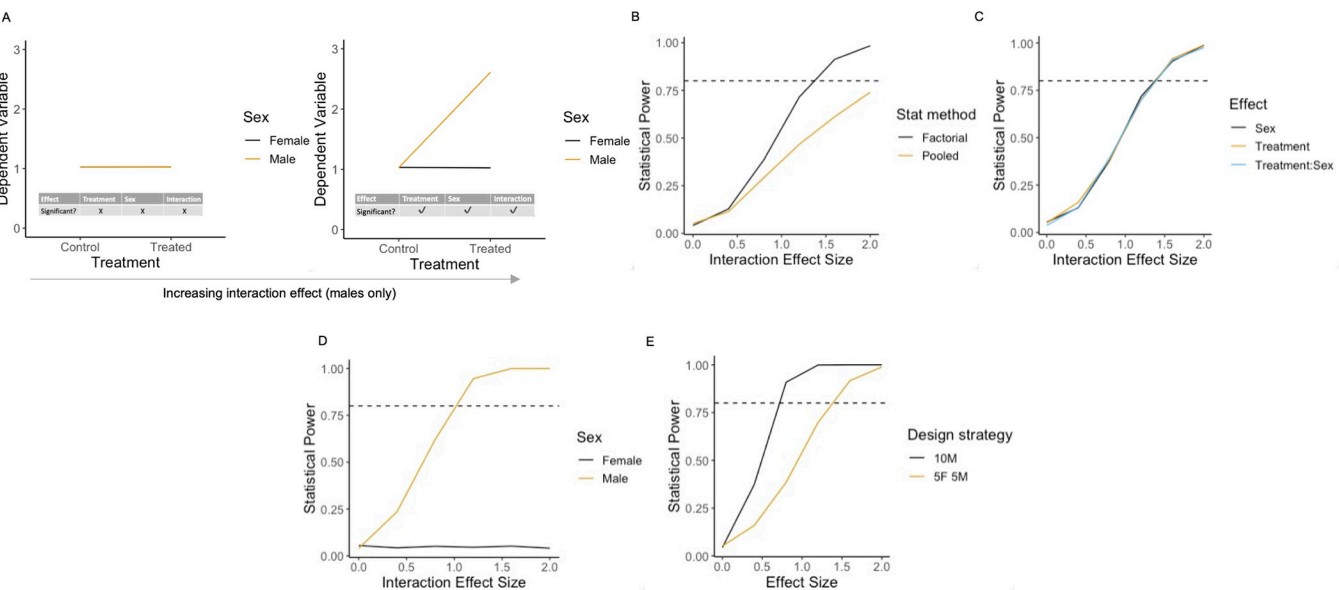

**Fig 5. The impact of a treatment effect specific to 1 sex on statistical power.** Results of simulations to calculate statistical power where there is an interaction driven by a treatment effect in 1 sex only. (A) Illustrative plots of constructed datasets, ranging from a zero to maximum treatment by sex interaction effect (0–2). (B) Statistical power to detect the main effect of treatment effect as a function of the interaction effect size for each statistical method. (C) Power for each model term within factorial analysis output. (D) Power for post hoc comparison control vs. treated within each sex. (E) Power for contrasting design strategies —10 males vs. 5 females and 5 males. Simulation $N = 1,000$ for each scenario assessed. For the graphs of power (B–E), the horizontal dashed line indicates the target power. The data underlying this figure can be found in https://doi.org/10.5281/zenodo.7806724.

## Scenario 3: Impact on statistical power when there is a treatment effect in 1 sex only

An extreme example of a treatment by sex interaction is when there is a treatment effect in 1 sex only (Fig 5A). As the interaction size increased, statistical power for the treatment effect was higher in the factorial pipeline compared to the pooled pipeline (Fig 5B). The lower power for the pooled pipeline arises as the between-sex variance was not accounted for. In this scenario, the power to detect the main effects of sex and treatment and the treatment by sex interaction were identical as the interaction effect size increased (Fig 5C). There was only power to detect a post hoc effect in the sex that exhibited the treatment effect, which increased as the interaction effect size increased (Fig 5D).

For this biological situation, where the treatment effect is unique to one sex, an additional simulation was run (Fig 5E) to compare the power when a single sex was used versus an inclusive design where the $N$ was split across the 2 sexes. The simulations confirm that there was a loss of power to using a sex-inclusive design compared to a design where the sex exhibiting the treatment effect was selected and studied in isolation (Fig 5E). In this rare scenario [21], we would argue that the conclusions obtained by studying only a single sex would be skewed. If the sex showing the treatment effect was included in the study, the researcher may have mistakenly assumed that the conclusion generalised to the entire population. Conversely, if the sex displaying low sensitivity to treatment were included, the researchers would have erroneously concluded that the treatment was ineffective.

In summary, when the treatment effect is unique to 1 sex, the simulations show there is a power benefit when a factorial pipeline is adopted compared to pooled pipeline. In comparison to a disaggregated strategy where the "right" sex had been selected, there is a reduction in power. However, our knowledge of the treatment effect is increased and unbiased when using an inclusive design with a factorial analysis.

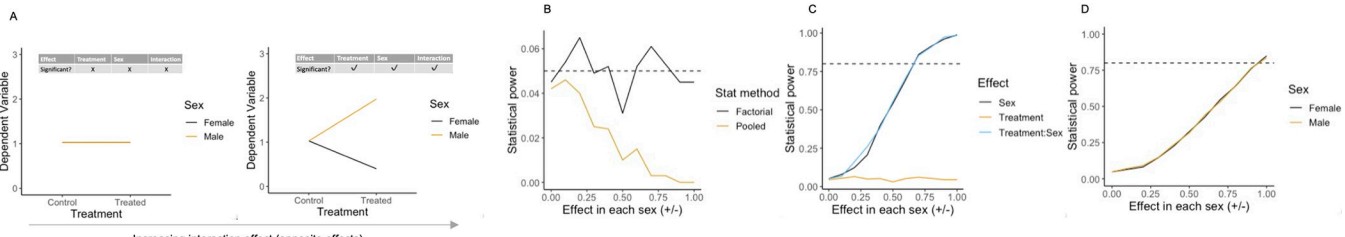

**Fig 6. The impact of an opposite treatment effect on statistical power.** Results of simulations to calculate statistical power where there is an interaction driven by opposite treatment effect in each sex. (A) Illustrative plots of constructed datasets, ranging from no effect to maximum opposite effect in each sex (0–1 male, 0 to −1 female). (B) Statistical power to detect the main effect of treatment as a function of the interaction effect size for each statistical method. (C) Power for each model term within factorial analysis output. (D) Power for post hoc comparison control vs. treated within each sex. Simulation $N$ = 1,000 for each scenario assessed. For the graphs of power (B–D), the horizontal dashed line indicates the target power. The data underlying this figure can be found in https://doi.org/10.5281/zenodo.7806724.

## Scenario 4: Impact on statistical power when there is an interaction driven by opposite sex effects

Though rare biologically [21], it is possible a treatment may produce an opposite effect between sexes (Fig 6A). For both the factorial and the pooled pipelines, the simulations show the statistical power for the main effect of treatment was low as expected (Fig 6B) because the mean effect of the treatment is zero as the treatment effect in each sex increases equivalently in opposite directions. The factorial analysis produced the expected false call rate (proportion of the time that a significant result is detected when there is no true difference) for the 0.05 significance threshold of approximately 5% for all effect sizes (Fig 6B). For the pooled pipeline, however, the simulations found that the false positive rate decreased as the size of the interaction effect increased (Fig 6B). The conservative behaviour of the pooling strategy arises from the influence of unpartitioned variance that is attributable to sex.

In the factorial analysis, power for the additional terms (sex and interaction) were equivalent and increased as the effect size increased (Fig 6C). In these situations, where you have a significant interaction but not main effect, the simulations demonstrate that the post hoc tests have high power in each sex (Fig 6D).

In summary, in this rare situation where there is no discernible main effect, but an interaction is significant, the post hoc tests are a crucial step to evaluate the effect. While the power is low for the main effect in this scenario, the power for the interaction is high which will trigger the correct interpretation. The pooled pipeline would fail to discern this critical biological insight. A disaggregated pipeline would have equivalent power but would not have a statistical test to support the conclusion that the effect depended on sex.

The power of the factorial design, where the $N$ is split across the 2 sexes is lower than a single sex design using the same $N$. However, the inclusion of only a single sex in this scenario would have resulted in a highly biased conclusion that should not be generalised to the broader population of interest. A strong opposite effect would merit follow-up studies to understand the basis of the differential treatment effect.

## Conclusions

The SABV philosophy is to include both sexes to allow a generalisable estimate with the potential to detect if there is a major difference in treatment response between sexes. This inclusive design, when combined with a factorial analysis, allows us to statistically test for whether the treatment effect depends on sex. This is driven by a desire to increase the translational confidence of the results and does not require experiments to be powered to detect treatment by sex

effects. Our simulations show that for most biologically expected situations (where treatment effects are similar across the sexes or there is a baseline sex difference), there is no need to increase the $N$ needed in the studies, rather the intended $N$ can be shared across the 2 sexes for a treatment. This strategy implies that power calculations for the treatment effect can be simplified to a 2-group comparison to estimate the total $N$ needed for a treatment, which is then shared between the 2 sexes. Alternatively, power calculations for factorial designs can be calculated using other methods (e.g., the *Superpower* package [25]). When there is a small difference in the size of the treatment by sex interaction, estimating the average effect is ideal for translational understanding as this is a generalisable conclusion. The simulations did demonstrate a loss of power when there is a large treatment by sex interaction (e.g., opposite or the effect only occurs in 1 sex). In this uncommon scenario and where the $N$ has been split across the 2 sexes, we may fail to detect a significant treatment effect (due to the lower power) but would gain important knowledge suggestive of a large sex dimorphism. Taken together, these conclusions support the recommendation to split the intended $N$ across the 2 sexes.

Our simulations additionally reveal the negative consequences of erroneously pooling or disaggregating the data by sex for analysis. For pooling, our results show that there is a loss of power to detect a treatment effect across most scenarios, including where there is a baseline sex difference and when there is a treatment by sex interaction. Pooling the data by sex also necessarily precludes identification of sex-specific effects, thus important biological knowledge would be lost in these scenarios. For disaggregation, when both sexes display a treatment effect, there is less power to detect it in each sex independently than via the main treatment effect term of a factorial analysis. Moreover, disaggregating precludes the detection of interaction effects, thus losing the ability to statistically assert a differential treatment by sex effect.

There are limitations of the simulations that we have carried out, and consequently, the conclusions we have reached in this manuscript. First, the simulations have been conducted on typical research scenarios aimed at determining a change in means between 2 groups and model continuous data with a normal distribution, equal variance, independent observations, and a balanced design. Where the goal of the study is not to test a mean difference in this context, our conclusions may not apply. The conclusions may not extend to a situation where the $n$ is extremely low, and this may also preclude a halving strategy (e.g., halving of 3 per treatment groups is unfeasible). Our investigations are also conducted in the context of a null hypothesis significance testing scenario where a decision is based on evaluating the $p$-value against a threshold (most typically $p < 0.05$). This is an area of significant ongoing discussion and debate [26]. However, given the current prevalence of $p$-values and their necessity for standard power calculations, we would argue that this limitation has minimal impact. Critically, the scope of this manuscript is limited to studies where the intention is to estimate a generalisable treatment effect, rather than exploring the dependence of a treatment effect on sex.

To facilitate progress in sex inclusion in in vivo research, it is crucial to provide scientists with both evidence and practical resources that challenge the barriers that currently stand in the way of change. This manuscript provides an in-depth analysis to explore the topic of power when both sexes are included to address the barrier that is the belief that inclusion requires an increase in the sample size. In this analysis, we have performed extensive statistical simulations to evaluate power under a range of common biological scenarios when splitting the $N$ across 2 sexes. Critically, we did not identify any common scenarios that result in a loss of power to detect treatment effects. Rarely, large interactions in the data may produce an appreciable decrease in treatment effect power. In these scenarios, we would argue the knowledge gained that the treatment has a differential impact between sexes outweighs the statistical loss of power. Furthermore, if a disease affects both sexes but the effect in the research model is observed in only one, this may bring into question the validity of the model or the

generalisability of the treatment. The simulations also demonstrate the pitfalls of some frequent analysis mistakes, including the inappropriate pooling and disaggregation of data collected from 2 sexes, which result in a loss of power to detect a treatment effect compared to a factorial analysis applied to the same data.

Additionally, we provide an example pipeline for analysing data collected from both sexes as a practical guide for scientists (S1 Case Study). The approaches above heavily depend upon the appropriate application of the correct factorial analysis methods, and it is therefore critical that laboratory scientists receive focused support in developing their statistical capabilities.

## Methods

### Ethics statement

All animal experiments were conducted in accordance with the United Kingdom Animal (Scientific Procedures) Act 1986 and associated guidelines, approved by institutional ethical review committees (Alderley Park Animal Welfare and Ethical Review Board; Babraham Institute Animal Welfare and Ethical Review Board) and conducted under the authority of the Home Office Project Licences (PF344F0A0). All animal facilities have been approved by the United Kingdom Home Office Licensing Authority and meet all current regulations and standards of the United Kingdom.

### Statistical simulations: Dataset construction

To explore the impact of sex either as a main effect (baseline sex differences) or when it interacts with the treatment on power, simulation studies were conducted. In the simulations, representative datasets with 5 animals per treatment group per sex were constructed by randomly sampling from a normal distribution after defining the mean and standard deviation of each treatment group. This process was repeated multiple times for each scenario of interest ($N = 1,000$) that enabled the subsequent evaluation of statistical power for each analysis pipeline of interest. We moved stepwise through 4 scenarios encompassing possible outcomes from studies testing a treatment effect in both sexes. Thus, the simulations differed by altering the specified means in each group (e.g., baseline sex difference, treatment by sex interaction) (see Table 1).

**Table 1.  Details of how the simulation datasets were constructed to represent various biological situations.**

| Simulation | Objective | Main effect of sex | Main effect of treatment | Interaction between the treatment and sex |
|---|---|---|---|---|
| 1 | **Impact of a baseline sex difference on statistical power** | **Varied** Signal added to means of males: none (0), small (0.5), or large (1) | **Varied** Signal added to the treatment group mean: 0–1 increments of 0.1 | **None** |
| 2 | **Impact of an interaction between sex and treatment on statistical power** | **None** | **Varied** Signal added to the treatment group mean 0–1 increments of 0.1 | **Varied** Signal added to the mean of the treated males None (0), small (0.5), or large (1) |
| 3 | **There is a treatment effect in 1 sex only** | **None** | **None** | **Varied** Signal added to males only: 0–2 in increments of 0.2 |
| 4 | **There is an interaction resulting from a treatment effect that is opposite across the sexes** | **None** | **None** | **Varied** Signal added to both sexes symmetrically in opposite direction Male: 0–1 increments of 0.1 Female: 0 to −1 increments of 0.1 |

Data was sampled from a normal distribution where the baseline characteristics were a mean of 1 and a variance of 0.5.

### Statistical simulations: Statistical analysis

The constructed datasets were statistically analysed either using a factorial pipeline or a pooled pipeline.

In the factorial pipeline, a regression analysis in R equivalent to a two-way ANOVA provided an assessment of the main effect of treatment, sex, and the interaction of treatment by sex. This was followed by a set of uncorrected post hoc pairwise tests between untreated and treated data in each sex using the R package *emmeans*. Sex is an unusual biological factor, as it depends on mendelian randomisation rather than experimenter randomisation in the allocation. The scenario we are considering, where both sex and treatment are included in the experimental design, may also be referred to as a stratified design, particularly in the clinical literature [27]. Here, we use the term factorial throughout as this is the typical terminology used by the biological community.

In the pooled pipeline, a Student's *t* test, after combining the data across the sexes for each treatment level, was conducted.

### Statistical simulations: Assessment of statistical power

The resulting datasets were analysed using both the factorial and pooled pipelines, and statistical power was defined as the proportion of the time a statistically significant effect was called for the model term of interest, at a significance threshold $p < 0.05$.

## Supporting information

**S1 Case Study. Example application of factorial analysis for data collected from 2 sexes.** This supplementary case study is intended as an example pipeline for analysing data from in vivo experiments collected from both sexes. It is not intended as an exhaustive tutorial, and there are other appropriate methods for analysing the type of data that we are presenting. (DOCX)

## Acknowledgments

We are grateful to Lorraine Miller for technical assistance with the experimental work and Chris Heath and Esther Pearl for helpful comments on earlier drafts of the manuscript.

## Author Contributions

**Conceptualization:** Benjamin Phillips, Natasha A. Karp.

**Data curation:** Timo N. Haschler.

**Formal analysis:** Benjamin Phillips, Natasha A. Karp.

**Project administration:** Natasha A. Karp.

**Visualization:** Benjamin Phillips.

**Writing – original draft:** Benjamin Phillips, Natasha A. Karp.

**Writing – review & editing:** Benjamin Phillips, Timo N. Haschler, Natasha A. Karp.

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
