## [Editor Report · Decision Letter 0]

9 Feb 2023

Dear Natasha, 

Thank you for submitting your manuscript entitled "Scientists should not increase the overall n by default when including both sexes in in vivo studies: evidence from statistical simulations" for consideration as a Meta-Research Article by PLOS Biology.

Your manuscript has now been evaluated by the PLOS Biology editorial staff, as well as by an academic editor with relevant expertise, and I'm writing to let you know that we would like to send your submission out for external peer review.

Once your full submission is complete, your paper will undergo a series of checks in preparation for peer review. After your manuscript has passed the checks it will be sent out for review. To provide the metadata for your submission, please Login to Editorial Manager (https://www.editorialmanager.com/pbiology) within two working days, i.e. by Feb 13 2023 11:59PM.

Kind regards,

Roli

Roland Roberts, PhD

Senior Editor

PLOS Biology

rroberts@plos.org

---

## [Decision Letter · Decision Letter 1]

22 Mar 2023

Dear Natasha,

Thank you for your patience while your manuscript "Scientists should not increase the overall n by default when including both sexes in in vivo studies: evidence from statistical simulations" was peer-reviewed at PLOS Biology. It has now been evaluated by the PLOS Biology editors, an Academic Editor with relevant expertise, and by three independent reviewers. 

Based on the reviews, we are likely to accept this manuscript for publication, provided you satisfactorily address the remaining points raised by the reviewers and the following data and other policy-related requests.

IMPORTANT - please address the following:

a) Please attend to the requests from the reviewers.

b) Regarding reviewer #1's comment "Perhaps another Article Type is more appropriate?," the answer is "yes" - I hadn't noticed that this was submitted as a regular Research Article; I have now switched it to a Meta-Research Article, which we feel should address this reviewer's valid concern - the advance here is in meta-research, not statistics.

c) Please change the Title to "Statistical simulations show that scientists need not increase overall sample size by default when including both sexes in in vivo studies"

d) Please address my Data Policy requests below; specifically, we need you to supply the numerical values underlying Figs 1BCD, 2BCD, 3BCDE, 4BCD, S1ABC, either as a supplementary data file or as a permanent DOI’d deposition. I note that you already have a Zenodo deposition, but access is currently restricted, so I can’t check compliance; please rectify this.

e) Please cite the location of the data clearly in all relevant main and supplementary Figure legends, e.g. “The data underlying this Figure can be found in S1 Data” or “The data underlying this Figure can be found in https://doi.org/10.5281/zenodo.7629021

We expect to receive your revised manuscript within two weeks. 

*Published Peer Review History*

*Press*

Sincerely,

Roli

Roland Roberts, PhD

Senior Editor,

rroberts@plos.org,

PLOS Biology

DATA POLICY:

Regardless of the method selected, please ensure that you provide the individual numerical values that underlie the summary data displayed in the following figure panels as they are essential for readers to assess your analysis and to reproduce it: Figs 1BCD, 2BCD, 3BCDE, 4BCD, S1ABC. NOTE: the numerical data provided should include all replicates AND the way in which the plotted mean and errors were derived (it should not present only the mean/average values).

DATA NOT SHOWN?

REVIEWERS' COMMENTS:

Reviewer #1:

Phillips et al. use simulations to show that including animals of both sexes and using an appropriate statistical analysis does not reduce the power to detect treatment effects, compared with using animals of one sex. The authors identify two misconceptions held by many scientists and three errors that are frequently observed in published papers. They then use simulations to demonstrate why the misconceptions are incorrect and provide better ways to analyse data.

This paper has been submitted as "Research Article" and I have been asked to comment on the "novelty and significance of the findings, its technical merit, the and experimental design". All the results are expected based on statistical theory and are well known to statisticians, and so are not novel or significant. However, demonstrating the results with simulations may be more convincing to many scientists. Having such a paper to reference when discussing these issues with colleagues and collaborators would be valuable. Perhaps another Article Type is more appropriate?

The statistical analyses are appropriate and the supplementary information provides a useful case study with real data. I cannot access the scripts and data on Zenodo as access is restricted.

I only have a few minor comments:

1) The font size is too small in all the figures and difficult to read. Adding

+ theme(text = element_text(size = 14))

to the plots should fix this.

2) Line 210: It would be useful to add Gelman and Stern's paper to the references here (https://www.tandfonline.com/doi/abs/10.1198/000313006X152649). I believe they were the first to draw attention to this problem.

3) Line 419-421: I think this sentence is missing some words. Is the intended meaning: "There is no requirement to perform a power calculation to detect …"?

4) Panel A in Figure 2 is referenced multiple times (lines 191, 193, 198, 200) but it's unclear where we should be looking as there are four graphs. Adding a more specific pointer would make it clearer (e.g. Fig 2A, top right).

Reviewer #2:

This is another outstanding and immensely practical contribution from this group. The paper does an excellent job in identifying the main issues, myths and all-too-common procedural errors involved in SABV studies. They provide transparent and readily reproducible methodology and code. I especially loved the explanatory boxes which made the methods clear without interrupting the flow of the text. 

Ethical oversight assurances are provided, and most animal-based procedures reported in accordance with ARRIVE 2.0. However, there was no statement as to the eventual disposition of the mice, and how they were euthanased prior to collecting kidney tissue samples. Please provide description of these methods. 

 A few other minor comments: 

1. The worked example is a really good idea. It could be made more practical for the average reader by streamlining the text quite a bit. The text is a bit clunky and focuses a lot on the results and less on the actual 'pipeline'. The text could be reduced by about 1/3rd & more bullet points added so it is easier to follow. Could the authors also consider placing this more upfront & incorporate something in the title & intro that states that the authors provide step by step methods on how to implement these features in actual studies (e.g. "tutorial" or "primer" into the title?) Or maybe consider a second companion paper? I am a bit worried that researchers who really need to see this will be put off by the heavy emphasis on the simulations & lessen the value of the practical implementation tips. 

2, Although the authors incorporated bias minimisation methods such as allocation concealment and randomisation in their own studies, there is almost no mention up front. These are essential components of experimental design. 

3. There is the added effect of cage. Was cage included as a block or were cage effects assumed to be zero? 

4. The sample size justification statement is far too vague to be verifiable. A few sentences could be added to explain. Not saying that it is, but as it stands, it seems to reinforce the all-too-common practice of choosing 5 per "group" because that is the usual per-cage stocking density. 

5. Mention should be made of sample size estimation in general for factorials. Powering factorials is different from the usual approach taken for two-group or single-factor completely randomised designs. As you know, main effects are estimated by comparing the means of 'sets of conditions' or a combination run, not by directly comparing the means of individual "groups". This needs to be mentioned upfront as well. Much of the resistance to adopting factorial designs is when researchers try to power them from G*Power which suggests far larger samples sizes are required. Sample sizes will also be too small for pairwise comparison of "groups". it is far more fruitful and less confusing to clarify that with factorials what is replicated are experimental runs (factor combinations) & that the researcher needs to think about total sample size, not group size. This is mentioned (sort of) in the text but it needs to be stated much more clearly. mention might be made of the Lakens and Caldwell paper (2021) on sample size for factorials. DOI:10.1177/2515245920951503

6. If the factorial is conducted for screening purposes, emphasis should be on how "relative promise" of the sex x trt interactions are evaluated e.g in terms of precision of the estimate. not statistical significance. mention of practical effect size could be added up front. 

Reviewer #3:

Summary: In this manuscript, Philips and Colleagues conduct statistical simulations to address common misconceptions surrounding the use of both sexes in in vivo studies. It is commonly thought that in order to design an experiment which is statistically powered to analyze data by sex, the sample size (n) must double. To address this, they conduct power analyses which mirror a variety of experimental conditions with varied effects of treatment and sex. They provide clear explanations surrounding the power analyses and the results. Their simulations highlight the fact that in most scenarios, when sexes are split across treatment groups, the sample size is enough to sufficiently power the study. 

Added Value of the Study: Several other articles have discussed how to design and analyze experiments involving both sexes. Most articles are written as reviews and provide overviews and/or tutorials of statistical methods. This manuscript expands upon that work by including the aforementioned power analyses and provides a case study which can be used for training purposes. It also reiterates the importance of factorial vs. pooled analyses, demonstrating that their incorrect use could lead to misleading results that ultimately detract from our understanding of how sex informs health and disease. 

Overall, I find this article to be well written and easily accessible to a broad audience. However, I acknowledge that I do not have expertise in statistical modeling and hope that other reviewers can comment on strengths and/or weaknesses that pertain to the methods used here.

---

## [Editor Report · Decision Letter 2]

18 Apr 2023

Dear Natasha,

Thank you for the submission of your revised Meta-Research Article "Statistical simulations show that scientists need not increase overall sample size by default when including both sexes in in vivo studies" for publication in PLOS Biology. On behalf of my colleagues and the Academic Editor, Marcus Munafò, I'm pleased to say that we can in principle accept your manuscript for publication, provided you address any remaining formatting and reporting issues. These will be detailed in an email you should receive within 2-3 business days from our colleagues in the journal operations team; no action is required from you until then. Please note that we will not be able to formally accept your manuscript and schedule it for publication until you have completed any requested changes.

Sincerely,

Roli

Senior Editor

PLOS Biology

rroberts@plos.org